# Label-Free Electrochemical Aptasensor for Sensitive Detection of Malachite Green Based on AuNPs/MWCNTs@TiO_2_ Nanocomposites

**DOI:** 10.3390/ijms241310594

**Published:** 2023-06-24

**Authors:** Zanlin Chen, Haiming Li, Miaojia Xie, Fengguang Zhao, Shuangyan Han

**Affiliations:** 1Guangdong Key Laboratory of Fermentation and Enzyme Engineering, School of Biology and Biological Engineering, South China University of Technology, Guangzhou 510006, China; 202210186526@mail.scut.edu.cn (Z.C.); 202221049169@mail.scut.edu.cn (H.L.); mjxie1111@163.com (M.X.); 2School of Light Industry and Engineering, South China University of Technology, Guangzhou 510006, China

**Keywords:** aptamer, cyclic voltammetry, potassium hexacyanoferrate, sensitive detection, differential pulse voltammetry

## Abstract

This study proposes a label-free aptamer biosensor for the sensitive detection of malachite green(MG) using gold nanoparticles/multi-walled carbon nanotubes @ titanium dioxide(AuNPs/MWCNTs@TiO_2_). The nanocomposite provides a large surface area and good electrical conductivity, improving current transfer and acting as a platform for aptamer immobilization. The aptamer and the complementary chain(cDNA) are paired by base complementary to form the recognition element and fixed on the AuNPs by sulfhydryl group, which was modified on the cDNA. Since DNA is negatively charged, the redox probe in the electrolyte is less exposed to the electrode surface under the repulsion of the negative charge, resulting in a low-electrical signal level. When MG is present, the aptamer is detached from the cDNA and binds to MG, the DNA on the electrode surface is reduced, and the rejection of the redox probe is weakened, which leads to an enhanced electrical signal and enables the detection of MG concentration by measuring the change in the electrical signal. Under the best experimental conditions, the sensor demonstrates a good linear relationship for the detection of MG from 0.01 to 1000 ng/mL, the limit of detection (LOD)is 8.68 pg/mL. This sensor is stable, specific, and reproducible, allowing for the detection of various small-molecule pollutants by changing the aptamer, providing an effective method for detecting small-molecule pollutants.

## 1. Introduction

Malachite green is a synthetic triphenylmethane dye that has bactericidal- and parasite-killing properties. It is usually used by fishmongers to prevent fungal infections and extend the life of fish with damaged scales [1]. Additionally, MG is a specific treatment for Saprolegnia fungus, a fish and fish eggs pathogen, making it widely used in commercial aquaculture [2]. Despite its benefits, studies have shown that MG is cytotoxic, mutagenic, and carcinogenic; even trace amounts can cause damage to the human immune and reproductive system [3,4]. Due to health concerns, MG has been prohibited from use in fish farming, and the European Union has set a maximum limit of 2 μg kg^−1^. Therefore, the rapid and accurate detection of MG in water is critical for formulating water-quality standards and assessing environmental risk levels to protect human health. Several methods have been developed for detecting MG, including high-performance liquid chromatography (HPLC) [5], mass spectrometry [6], HPLC-mass spectrometry [7], immunological assays [8], surface-enhanced Raman scattering [9], and molecularly imprinted polymer [10]. While these methods are accurate and sensitive, they often require sophisticated instruments and trained staff. As a result, developing a simple method for rapidly detecting MG in non-laboratory conditions is necessary.

Aptamer is an artificial, single-stranded oligomer probe of DNA and RNA composed of 10–100 bases obtained by the systematic evolution of ligands by exponential enrichment (SELEX) [11]. This can fold into distinct secondary structures and bind to target substances such as bacteria, heavy metal ions, proteins, small molecules, etc. [12,13,14]. Due to the high chemical stability and simple operation of aptamers, aptamer-based biosensors have received much attention from researchers. Generally, the signal label is modified on the aptamer, and the signal strength transmitted by the label changes according to the conformation of the aptamer, thus converting the conformation change into visible information. Based on the different types of visual signals, sensors can be divided into colorimetric sensors, fluorescence sensors, and electrochemical sensors. The electrochemical aptamer sensor (E-apt sensor) is an important type of aptamer sensor that combines the sensitivity of electrochemical detection with the specificity of aptamer [15]. Over the years, E-apt sensors have rapidly developed and been applied to detect a variety of substances, including mercury ions [16], Bisphenol A [17], tumor cells [18], etc.

E-apt sensors possess a higher sensitivity, which allow them to detect targets at 10^−6^ nM levels. In addition to the binding activity of the aptamer, the high sensitivity of E-apt sensor is due to the intelligent use of nanomaterials. Electroactive nanomaterials usually have high-electrical conductivity and a large specific surface area, allowing them to transform and amplify signals [19]. The use of immobilized aptamers on the surface of nanomaterials through intermolecular forces has been observed to significantly improve the specificity and sensitivity of sensors [20]. Among carbon nanomaterials, MWCNTs stand out due to their unique electrical transport properties, large specific surface area, and excellent chemical, mechanical, and thermal stability. Pinar Kara et al. [21]. developed an aptamer-based MWCNT biosensor in 2010; MWCNTs were used as modifiers of screen-printed carbon electro transducers (SPCEs), demonstrating improved characteristics compared to the bare SPCEs. Samira Yazdanparast et al. [22] directed research toward the application of a nanocomposite containing PGA and MWCNT to modify the surface of glassy carbon electrode (GCE). They observed that a carbon nanotube-PGA composite provides durability and activity with a large surface area, which can facilitate the electron-transfer reaction. Since then, various sensors based on multi-walled carbon nanotubes have been developed, like Multiwalled carbon nanotubes with graphene oxide [23], core-shell nanofibers [24], conducted polymer poly-3,4 ethylenedioxythiophene [25], etc.

For the past few years, many semiconductor materials have been used to enhance electrochemical sensing signals. Andrei Pligovka et al. [26,27] determined that randomizing two-level 3D column-like nanofilms hexagonally through porous anodic alumina can effectively enhance the current signal. TiO_2_ is a semiconductor material with strong adhesion and stable properties. Its combination with MWCNTs can enhance the electrochemical performance of MWCNTs and demonstrate a high sensitivity to target substances in detection [28]. Meanwhile, nanomaterial AuNPs have also been proven to effectively enhance the electrochemical signal and further improve the detection ability of the sensor. Moreover, AuNPs can firmly bind to the thiol aptamer through the Au–S bond and play the role of fixing the aptamer [29]. Three aptamers with excellent performance in detecting MG were identified in our previous work. They constructed a fluorescence sensor based on those aptamers, with the LOD being 1.82 ng/mL [30]. However, this sensor requires the fluorescent modification of the aptamer, which had a relatively low recovery rate for samples with lower concentrations.

To address these limitations and improve the sensitivity, convenience, and precision of MG detection, an E-apt sensor based on a gold electrode (AuE) modified by AuNPs/MWCNTs@TiO_2_ was constructed. The MWCNTs@TiO_2_ exhibited good charge-transfer properties, and the AuNPs with strong conductivity are deposited on the MWCNTs to obtain better electrochemical performance. This composite nanomaterial can significantly improve the conductivity of the sensor, which is also the key factor for the sensor to detect MG with high sensitivity. The aptamer forms a recognition element with its cDNA, and the sulfhydryl group on the cDNA forms an Au–S bond with theAuNPs, which is fixed on the electrode surface. Since the DNA itself is negatively charged, it repels the redox probes, resulting in fewer redox probes contact with the electrode surface, resulting in a decrease in the current signal. When MG was added, the aptamer binds to MG, changes its conformation, and falls off cDNA, allowing for more redox probes to reach the electrode surface. Resulting in a significant increase in the current signal. The rise of the current has an obvious linear relationship with the concentration of MG, so the concentration of MG can be judged according to the rise degree of the current. The specific sensor works as shown in Figure 1. The sensor has a low detection limit and a large detection range, and the nanomaterial used to modify the electrode is simple to produce. Therefore, this sensor has great potential for detecting malachite green in aquatic products. The current signal changes on the gold electrode were detected by the differential pulse voltammetry (DPV) method to determine the presence of MG.

## 2. Results

### 2.1. The Characterization of MWCNTs@TiO_2_

TiO_2_, MWCNTs and MWCNTs@TiO_2_ were dissolved in anhydrous ethanol, respectively (Figure 1A). The above solutions were titrated on the surface of the gold electrode with 2 μL separately and dried at room temperature for 30 min. The morphology and structure of the prepared nanomaterials were characterized by scanning electron microscopy (SEM). Figure 1B shows the microscopic morphology of the bare gold electrode under SEM. The plotting scale of observed TiO_2_ and MWCNTs was adjusted to 100 nm. TiO_2_ could be observed to be stacked together in a uniform spherical shape (Figure 1C). According to Figure 1D, it was observed that MWCNTs showed tubular structures and were intertwined with each other, indicating that the microstructure of MWCNTs and TiO_2_ was completely different. As shown in Figure 1E, plenty of spheres attached to tubes of carbon nanoparticles can be observed by enlarging the scale to 500 nm. This indicates that TiO_2_ was successfully mixed with MWCNTs to form MWCNTs@TiO_2_. The EDS scanning region of the MWCNTs@TiO_2_ and the scanning results of titanium (Ti), carbon (C), and oxygen(O) elements were displayed in Figure 1F. All of the elements were evenly distributed on the surface, without an obvious element enrichment phenomenon. Oxygen atoms accounted for the highest proportion of 42.23% in the mixture, carbon atoms accounted for a slightly lower proportion of 38.37%, and titanium atoms accounted for less than 20%. These results indicate that MWCNTs@TiO_2_ nanomaterials were successfully fabricated.

### 2.2. The Characterization of E-Apt Sensor Fabrication

Cyclic voltammetry (CV) is a conventional electrochemical method for detecting changes in current signal that is easy to operate and analyze. As shown in Figure 2, a pair of good reversible redox peaks can be observed in the unmodified gold electrode (Curve A), demonstrating that [Fe (CN)_6_]^3−/4−^ has good redox properties in this system. Subsequently applying MWCNTs@TiO_2_ drops to the surface of the gold electrode (Curve B). Since the composite material can enhance the electrical conduction efficiency, the modified electrode was significantly enhanced compared with the bare gold electrode. After the addition of AuNPs, the peak current was further increased due to the high electrical conductivity of AuNPs. The thiolate cDNA1 and cDNA2 were firmly bound to AuNPs via an Au–S bond [31], and aptamers were immobilized to cDNA via base complementary pairing, resulting in the formation of a dense molecular layer. The molecular layer forms a barrier on the surface of the modified electrode, preventing the [Fe (CN)_6_]^3−/4−^ in the electrolyte from reaching the electrode surface, resulting in a significant decrease in the current (Curve C). Then, 6-mercapto-1-hexanol (MCH) was added to block non-specific binding sites and prevent non-specific binding of MG on the surface of the modified electrode (Curve D). This results in further reduction of the active sites on the electrode surface that can contact the redox probe, and a further decline in the current signal Aptamer was detached from cDNAs and formed a complex with MG in the presence of MG. The barrier blocking [Fe (CN)_6_]^3−/4−^ was broken, and [Fe (CN)_6_]^3−/4−^ could freely contact the electrode surface, while the current intensity increased significantly (Curve E). In addition, the degree of increase was positively correlated with MG concentration. Use the DPV method to verify the current changes in the above experimental steps. The current reaction was simply divided into three stages: “increase–decrease–increase.” The trend was exactly parallel to what CV had observed (Figure 3). Especially in the current rebound stage after the appearance of malachite green, the results of DPV detection also show that the rising current intensity is proportional to the malachite green concentration. As a result, an E-apt sensor for sensitive detection of MG was successfully established.

### 2.3. Optimization of Experimental Conditions

This chapter explores the appropriate conditions to prepare aptamer sensors to achieve optimal detection results. All experiments were carried out at room temperature with K_3_[Fe (CN)_6_] electrolyte, and CHI660E was used to detect changes in electrical signals. The fixation of the aptamer is a key step to affect the detection effect of the E-apt sensor. This is because the nature of the sensor is to express the conformational change of the aptamer with visible information, so the fixation of the aptamer as a recognition element is particularly important. The fixation effect is determined by the vitality, concentration, and incubation time of aptamer. Aptamers are essentially nucleic acid molecules, and the pH of the environment has a great influence on their activity. To obtain high-activity aptamers, the pH of the buffer solution is very important. In addition to vitality, concentration also plays an important role. When the concentration is too low, the number of aptamers fixed on the electrode surface is insufficient to bind with trace MG, and the detection effect is poor. When the concentration is too high, even if MG binds to the aptamer, there will still be a large amount of double-stranded structure on the electrode surface, unable to react to [Fe (CN)_6_]^3−/4−^ probe and detect trace MG. Except for concentration, incubation time is a key factor. If the incubation time is too short, the aptamer cannot be well-combined on the electrode surface. If it is too long, the assembly efficiency will be affected. The same principle applies to the optimization of incubation time for MCH.

The optimal vitality, concentration, and incubation time of aptamer were investigated. The influence of MCH incubation time was also investigated. The difference between the current after incubation (I_A_) and the current before incubation (I_B_) was used as a determiner (ΔI, I_B_^−^ I_A_). As shown in Figure 4A, with the increase of pH, ΔI rose rapidly originally, reached a peak when pH reached 7.5, and then began declining on a smaller scale. This indicates that the aptamer activity is highest in the buffer with a pH of 7.5. DNA repels [Fe (CN)_6_]^3−/4−^-probes because it contains negatively charged phosphoric acid residues. As a result, ΔI increases with the increase of aptamer concentration to reach the maximum at 2 μM and does not change with the concentration. Furthermore, the 2-μM aptamer can completely cover the electrode surface, and there is no need to incubate with higher concentrations of aptamers (Figure 4B). Therefore, 2 μM was selected as the optimal aptamer incubation concentration, and a time of 10 min was selected as the time optimization interval in the optimization process. As shown in Figure 4C, the current change value is small when the incubation time is only 10 min, but when the incubation time reaches 30 min or longer, the current change reaches its maximum and remains constant. As a result, the best incubation time was determined to be 30 min. The optimization of MCH incubation time was in the same situation (Figure 4D).

### 2.4. Analytical Performance of E-Apt Sensor

#### 2.4.1. Linear Response and LOD for MG

The detection of MG was performed in optimized experimental conditions following the fabrication of the E-apt sensor. Then, 2 μL of different concentrations of MG (0, 0.01, 0.1, 1, 10, 100, 1000 and 10,000 ng/mL) were titrated on the surface of the electrode and detected electrical signal after natural drying. The high affinity of MG and aptamer coupled them into a complex, destroying the double-stranded structure of aptamer and cDNA. The redox probe could make contact with the electrode; as a result, the current response increased. It was evident from Figure 5A that the degree of current increase is proportional to the concentration of MG, and that the relationship is linear in the concentration range of 0.01 ng/mL to 1 μg/mL. In addition, LOD is 8.68 pg/mL. The linear regression equation was calculated as y = 51.0585 + 3.8892x, R^2^ = 0.98313 (Figure 5B). In addition, the performance of the proposed sensor is compared to other MG sensors published in recent years (Table 1). The results demonstrate that the proposed sensor has a broad detection range and a good limit of detection.

#### 2.4.2. Specificity, Reproducibility, and Stability of Proposed E-Apt Sensor

In the process of preparing an accurate and reliable E-apt sensor, the importance of specificity is the same as sensitivity. The specificity of the sensor is detected by adding different kinds of small molecular agricultural and veterinary drugs under the same condition. Acephate (ACE), tricyclazole (TRI), thiamethoxam (THI), and glyphosate (GLY) were compared with MG by 50-fold concentrations, respectively. As shown in Figure 6A, even at a 50-fold concentration (5 μg/mL), the sensor’s response to other drugs was only about half that of MG, which currently contrasts with blank. This phenomenon demonstrates that the sensor has high specificity.

In addition to specificity, reproducibility is also a very important indicator for evaluating whether the sensor is stable and reliable. The experiment was repeated three times on four different AuEs. The largest RSD value of the three time experiments was only 2.2% (Figure 6B). Meanwhile, the prepared sensor was stored at 4 °C for 1, 3, 5, and 7 days, respectively, to test its stability. The detection ability of MG was compared to that of the newly prepared sensor, as shown in Figure 6C. When the storage time is less than five days, the recovery rate for MG remains stable at about 97%. The recovery rate of the sensors stored for seven days could also reach 90%. In addition, the experimental water samples from different sources, and whether they will affect the detection effect of the sensor, are verified. The results show that the sensor can still possess good stability in different water samples (Figure 6D). This indicates that the sensor has a high specificity, reproducibility, and stability. The recovery rate of the E-apt sensor for MG at different concentrations in the two water samples was listed in Table 2. In Fishery water, the recoveries were 94.8–102.4%, with RSD less than 1.79%. In tap water, the recovery rate was 92.8–100.7%, and the RSD value was no more than 3.19%. Even the concentration of MG was only 0.01 ng/mL; the sensor had a recovery rate of more than 90%. The appearance of E-apt sensor was shown in Figure 7.

## 3. Discussion

Malachite green is prohibited for use in fish feed due to its toxicity and oncogenicity. However, illegal traders still use it as a drug to treat saprolegniasis, posing serious threats to water environment quality and consumer safety. In this study, we developed a label-free E-apt sensor based on AuNPs/MWCNTs@TiO_2_ for ultrasensitive detection of MG in fishery water. The sensor is simple to manufacture, easy to operate, and has excellent sensitivity, specificity, stability, and reproducibility. It provides an effective method for the detection of other small molecular pollutants by changing the aptamer to achieve the efficient detection of the corresponding target. Although this sensor shows promising prospects for wide applications in food safety and environmental monitoring, the detection equipment is inconvenient to carry around, making on-site monitoring challenging. Therefore, future research should focus on the development of miniaturized, intelligent, detection devices that can be combined with mobile phones and other electronic devices to achieve portable, on-site detection.

## 4. Materials and Methods

Materials and chemicals: Malachite green oxalate (MG, 98%, CAS: 2437-29-8) was purchased from the China National Institute of Metrology Co., Ltd. (Beijing, China). 6-mercapto-1-hexanol (MCH, 98%, CAS: 1633-78-9), multi-walled carbon nanotubes (MWCNTs, ≥98%, CAS: 308068-56-6), titanium dioxide (TiO_2_, 99%, CAS: 13463-67-7), Gold (III) chloride hydrate (Au ≥ 47.5%, CAS: 27988-77-8) was obtained from Aladdin Biochemical Technology Co., Ltd. (Shanghai, China). Potassium hexacyanoferrate (III) (K_3_[Fe (CN)_6_], 99.5%, CAS: 13746-66-2) was purchased from Macklin Biochemical Co., Ltd. (Shanghai, China). Potassium chloride (KCl, 99%, CAS: 7447-40-7) was purchased from Damao Chemical Reagent Factory (Tianjin, China). Anti-MG aptamer (Apt) with the sequence of 5′-CCA TGC GAC GGA CAG CAC GTG TCA CCG CGA TCA GCC-3′, the corresponding cDNA1 sequence of 5′-SHC_6_-TTT TTG GCT GAT C-3′, and the corresponding cDNA2 sequence of 5′-TCG CAT GGT TTT T-SHC_6_-3′ were synthesized and purified by Sangon Biotech. Co., Ltd. (Shanghai, China). DPBS buffer was prepared with sterilization ultrapure water (137 mM NaCl, 2.7 mM KCl, 1.5 mM KH_2_PO_4_, 8 mM Na_2_HPO_4_, 1 mM CaCl_2_, 0.5 mM MgCl_2_, and 1 L, pH = 7.4) was used for assembling the aptasensor. All analytical pure chemical reagents were purchased from Damao Chemical Reagent Factory (Tianjin, China). Ultrapure water was applied throughout the experiments.

Apparatus: The scanning electron microscopy (SEM) and the energy dispersive spectroscopy (EDS) images were collected with Merlin (Zeiss, Oberkochen, Germany). The electrochemical measurements were performed by CHI 660E electrochemical workstation (Shanghai Chenhua Instrument Corporation, Shanghai, China). The three-electrode system consisted of working AuE (diameter 3 mm), an Ag/AgCl electrode as a reference electrode, and platinum wire as an auxiliary electrode (Tianjin Incole Union Technology, Tianjin, China). Vortex mixer uses Vortex-Genie 2 (Scientific Industries, Bohemia, NY, USA). All electrochemical measurements were carried out at room temperature. All aqueous solutions were prepared with ultrapure water by a water purification system (PALL, Port Washington, NY, USA, 18.2 MΩ·cm at 25 °C).

Preparation of MWCNTs@TiO_2_ compound: MWCNTs (100 mg) and TiO_2_ (50 mg) were dissolved in 25 mL of absolute ethanol and placed on a vortex mixer with strong shaking for 3 min, followed by 60 min of ultrasonic treatment until obtaining a highly dispersed grey-black solution, indicating that the MWCNTs@TiO_2_ compound was obtained.

Fabrication of E-apt sensor: Before being modified, AuE was immersed in piranha solution (concentrated sulfuric acid: hydrogen peroxide = 7:3) for 10 min, rinsed with ultrapure water, and polished with 1.0 μm of alumina powder to mirror the surface. The AuE was then ultrasonically treated in ethanol and ultrapure water for 1 min and dried with a nitrogen stream. First, 2 μL of MWCNTs@TiO_2_ was dripped onto the surface of the AuE and dried naturally at room temperature. Then, under the same conditions, 2 μL of AuNPs droplets were applied to the preliminarily modified AuE. After that, 2 μL of Aptmix, which was obtained by mixing 1 mL of 8 μM cDNA1, 1 mL of 8 μM cDNA2, and 2 mL of 4 μM Apt overnight incubation, was titrated on the surface of the modified coating and naturally dried. A compact DNA monolayer is formed on the surface of the coating by the Au–S bond and π–π bond. Finally, 2 μL of 1 mM MCH were used to block non-specific binding. The modified electrode was incubated with different concentrations of MG solution, and the change of the current was detected by the DPV method. The assembly process and the detection principle of the proposed E-apt sensor were shown in Figure 1.

Sample preparation: The actual water samples were purchased from local aquatic products markets (Guangzhou, China) and used for the detection of MG. The actual water samples were subjected to simple pretreatments, such as standing, filtration with 0.22-μm film, and 5-fold dilution with DPBS to remove solid impurities before spiking with different concentrations of MG standard samples (0, 0.01, 0.1, 1, 10, 100, 1000, and 10,000 ng/mL).

Electrochemical detection: The gold electrode modified with nanomaterials was used as the working electrode, an Ag/AgCl electrode as a reference electrode, and a platinum wire as an auxiliary electrode. CV and DPV measurements were performed in 0.1 M KCl solution containing 5.0 mM [Fe (CN)_6_]^3−/4−^. The scan rate of CV was set to 100 mV s^−1^, while the potential range was set between −0.4 V and 0.8 V to detect signal changes. Ten cycles were scanned to obtain a stable CV curve. The DPV data were taken in the same solution from −0.1 to 0.4 V, pulse time of 50 ms, and amplitude of 0.05 V.

## 5. Conclusions

In this study, we developed a label-free E-apt sensor based on AuNPs/MWCNTs@TiO_2_ for ultrasensitive detection of MG. The signal of the electrode modified by nanomaterials can be amplified by as much as four times, which is convenient for sensitive detection of MG and achieve ultra-micro measurement. The nanomaterial has the advantages of simple preparation, low cost, and wide application potential. The sensor demonstrates a good linear relationship for the detection of MG from 0.01 to 1000 ng/mL, and the detection of MG can be completed within 30 min, satisfying the requirements for rapid monitoring. In addition to sensitivity, stability and reproducibility are equally guaranteed. The sensors were prepared with different gold electrodes and applied to MG detection in different water samples. Results showed that the stability and reproducibility of the sensor are excellent. Relative to other properties, the specificity of this sensor may not be so good. It is believed that this is due to the limitations of the aptamer itself, but the sensor can be well-used in the detection of other agricultural and veterinary drugs by changing the recognition element, provide some references for the construction of other sensors.

## Data Availability

There is no electronic datasheet associated with this paper. No data in electronic repository.

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
