# Peer review of "Label-Free Electrochemical Aptasensor for Sensitive Detection of Malachite Green Based on AuNPs/MWCNTs@TiO2 Nanocomposites"

_ijms, 2023, doi:10.3390/ijms241310594_

Round 1
Reviewer 1 Report
1. Were the DPV traces in Fig. 4A the response of one same electrode to different MG concentrations? If one same sensor was used to measure multiple MG concentrations, what method was used to regenerate the electrode? How reliable was the "cleaning" method? If the traces were the response of different electrodes, it is hard to justify that the change of current was from the variation of different electrodes or resulted from MG concentration changes.
2. It is great that the authors have aimed to develop a highly sensitive sensor for MG detection, however it’s unclear if it’s really required. While the authors mention in their introduction the maximum limit allowed for MG, can the authors provide any information on what kind of minimum concentrations are observed in real samples? If the samples never see such low concentrations, the sensor sensitivity is not really required. Instead, the sensor specificity should be worked at more because that doesn’t seem to be very promising from Fig. 5A.
3. The authors should test their sensors at much higher concentrations of MG as well, to show the saturation point of the sensor.
4. Fig. 5C, under what conditions are the sensors stored for the stability test? This should be mentioned somewhere in the manuscript.
English is fine.
Reviewer 2 Report
The authors have submitted a manuscript entitled "High Sensitivity Screen Printed Non-Enzymatic H2O2 Sensor Based on Porous Cobalt Hexacyanoferrate Nanospheres" and a reported a method for the synthesis of a label-free aptamer biosensor for the sensitive detection of malachite green (MG) using gold nanoparticles/multi-walled carbon nanotubes @ titanium dioxide.
Important grammatical errors and punctuation mistakes can be found in the entire manuscript, being several times nearly impossible to understand the content. English need revision overall.
I consider this manuscript is not suitable for publication in International Journal of Molecular Sciences.
Some suggestions:
1_Check the use of punctuation and capital letters.
2_ Line 61: Liu et al. (insert reference)
3_Figure 1. It doesn't make sense, it's a black image with a scale. What are the authors trying to show?
4_Line 90, instead of “Various nanomaterials” it is better to mention the specific “nanomaterials.”
5_ Figures 1B-1D are not even referenced in the text.
6_Section 2.2. Page 3. This section contains several punctuation errors.
7_Figure 2. The figure is confusing. I recommend showing a table with the peak potentials to be able to analyze if the electrical transfer has really improved or not, since, although the current intensity increases, we see that the authors have not considered that the intensity of the double layer also increases.
8_Line 116-177. “The thiolate cDNA1 and cDNA2 were firmly bound to AuNPs via an 116 Au-S bond”. It should be proven or, at the very least, authors should consider including a reference to support this finding.
9_Line 121. “MCH” has not been previously defined.
10_Fig. 3B and Fig. 3D are not referenced in the text.
11_Fig 3. It should be indicated under what conditions these measures have been carried out, especially the media and the experimental set up in which they have been conducted.
12_Section 2.4.2. Specificity, reproducibility, and stability of proposed E-apt sensor
The authors should explain why these possible interferents have been chosen.
The way the article has been written makes it unclear and presents unconnected and sometimes difficult-to-follow arguments. The language is not well organized, and many spelling or format mistakes can be found in the article. There is a great space to improve this work.
Reviewer 3 Report
Manuscript ijms-2437499
Dr. Shuangyan Han manuscript is devoted to a label-free aptamer biosensor for the sensitive detection of malachite green using AuNPs/MWCNTs@TiO2. The AuNPs/MWCNTs@TiO2 provide a large surface area and good electrical conductivity, improving current transfer and acting as a platform for aptamers immobilization. This work combines complex lab research from two science remote areas – nanomaterials and biomedicine. It should be noted that a lot of lab work has been done. However, in this form the manuscript cannot be published and needs to be improved:
1. The authors claim to create a biosensor, but the manuscript does not provide a detailed description of its design and manufacturing process. The manuscript presents laboratory techniques and sensitive material. At this stage of research, it is too early to talk about a ready-made biosensor.
2. The work is full of pronouns. Please, rewrite sentences with pronouns. Personal pronouns aren't used in scientific works.
3. The keywords repeat words from the title and abstract. Search engines when searching index not only keywords, but also words from the abstract and title.
4. The introduction is pretty well written. However, the authors did not mention other semiconductor nanostructured metal oxides that can be used instead of titanium dioxide for immobilization, such as niobium oxide. The authors should note in the introduction the work on the formation of a biosensor 10.3390/ma16030993 based on niobium oxide and the study of its semiconductor characteristics 10.1109/TNANO.2019.2930901.
5. The figs are not design according to the rules of MDPI. Authors should study the requirements better. For example, why is there a white box on the left? Figures should be grouped more compactly. Make the fonts and the size of sub-caption the same in all figs.
6. Why is a scale up to 11.5 presented in Fig 1F if there are no more peaks after 5?
7. Line 109. Readers should be able to guess what «CV» is?
8. Fig 2 is difficult to understand. The explanations are hard to read because of the small print. It is not clear which curves refer to which curves. In the case of black and white printing of the article it will be impossible to identify the curves.
9. Line 171. MG? LOD? It is not recommended to enter abbreviations in the abstract.
10. Figure 5 is of poor quality, poorly readable.
11. Authors should present a picture of the sensor itself, not just the materials from which it is made. Ideally, we would like to see a photo of the sensor, what it looks like, and a SEM image of the sensor structure pie.
12. In general, the work is quite interesting and worthy of attention. However, it is quite hard to read, there are many unclear points. The authors should present this material more clearly and distinctly.
13. The introduction does not clearly articulate the purpose of the work.
14. The methodological part and the results are reversed.
15. The reference is not made under the rules of MDPI.
16. This is the first time I've seen a manuscript without a conclusion. This alone deserves to reject the manuscript.
The manuscript is full of pronouns. Should rewrite sentences with pronouns. Personal pronouns aren't used in scientific works.
Round 2
Reviewer 2 Report
The paper has been improved in general.
The paper has been improved in general.
Author Response
There are no comments of reviewer 2.
Reviewer 3 Report
Manuscript ijms-2437499 Round 2
Dr. Shuangyan Han manuscript is devoted to a label-free aptamer biosensor for the sensitive detection of malachite green using AuNPs/MWCNTs@TiO2. The AuNPs/MWCNTs@TiO2 provide a large surface area and good electrical conductivity, improving current transfer and acting as a platform for aptamers immobilization. This work combines complex lab research from two science remote areas – nanomaterials and biomedicine. It should be noted that a lot of lab work has been done.
Author’s response 1:
Dear reviewers, thanks for your comments. According to your comments, we modified the keywords to avoid excessive overlap with the title and summary. The new keywords are aptamer, electrochemical, malachite green, sensitive detection, and differential pulse voltammetry.
Comment 1:
It is better to remove all words from the title and abstract. It doesn't make sense. The system indexes keywords along with the title and abstract, you just deprive yourself of new citations and hits.
Author’s response 2:
Dear reviewers, we have made changes to the corresponding images and added subheadings according to MDPI specifications. All images have been placed in the center and are about the same size.
Comment 2:
It's much better now. However, the figs are still of poor quality. The authors need to work more on the design of the graphic material. The pictures are the face of an article, it should be like candy to make you want to read the article.
Author’s response 3:
Dear reviewers, Because the main part of the sensor is a gold disk electrode, its structure is relatively simple, and we are worried that simply placing a photo of the sensor is not beautiful enough. The general appearance of the sensor is shown in the lower right corner of Figure 1E.

Comment 3:
That's cool. You have real sensors. That makes a difference. You should make that a separate fig at the end of the manuscript. That's your main result, you have to stick it out. Unfortunately, the little humble endnote in fig 1E doesn't draw attention to itself. Make it a separate fig.
Athor’s response 4:
Dear reviewers, this is because the journal's requirements for conclusions are not mandatory, conclusion be added to the manuscript just the discussion is unusually long or complex. According to your suggestion, we have decided to make a supplementary statement to the conclusion.
In this study, developed a label-free E-apt sensor based on AuNPs/MWCNTs@TiO2 for ultrasensitive detection of MG. The large specific surface area and high electrical conductivity of the nanocomposites were utilized to enhance the initial cur-rent value, amplify the effect of MG, and achieve ultra-micro measurement. The nanomaterial has the advantages of simple preparation, low cost and wide application potential. The sensor is convenient to manufacture, easy to operate, and has excellent sensitivity, sta-bility, and reproducibility. It can complete the detection of MG within 30 minutes, sat-isfying the requirements for rapid monitoring. Relative to other properties, the speci-ficity of this sensor may not be so good, suspect that this is due to the limitations of the aptamer itself, but the sensor can be well used in the detection of other agricultural and veterinary drugs by changing the recognition element, provide some references for the construction of other sensors.
Comment 4:
The authors should improve the conclusion. It is good that it appeared. The conclusion should consist of two parts. In the first, the authors should summarize what was done in the paper and note the general result obtained with numerical values. After that, point by point list the main results with numerical values, on the basis of which the main result was obtained and the purpose of the work was achieved.
The conclusion is an extremely important part and it should be exhaustive and self-sufficient, that is, read separately from the main text. It is as important as, and related to, the title, abstract, and purpose of the paper. There can be no article without a conclusion. As well as without the purpose of the paper and the title. Please don't forget that.
Comment 5:
I strongly recommend that authors master the review mode in MS Word. This will simplify the work of the editor and reviewer.
Good luck, goodbye!
